# Sesquiterpenes dominate monoterpenes in Northern wetland emissions

Heidi Hellén[1], Simon Schallhart[1], Arnaud P. Praplan[1], Toni Tykkä[1], Mika Aurela[1], Annalea Lohila[1,2], Hannele Hakola[1]

[1]Finnish Meteorological Institute, P.O. Box 503, 00101 Helsinki, Finland

[2]Institute for Atmospheric and Earth System Research / Physics, Faculty of Science, University of Helsinki, Finland

*Correspondence:* Heidi Hellén (heidi.hellen@fmi.fi)

**Abstract.** We have studied biogenic VOC emissions and their ambient concentrations at a sub-Arctic wetland (Lompolojänkkä, Finland), which is an open, nutrient-rich sedge fen, and a part of the Pallas-Sodankylä Global Atmosphere Watch (GAW) station. Measurements were conducted during the growing season in 2018 using an in situ thermal desorption - gas chromatograph - mass spectrometer (TD-GC-MS).

Earlier studies have shown that isoprene is emitted from boreal wetlands and it turned out to be the most abundant compound in the current study also. Monoterpene (MT) emissions were generally less than 10% of the isoprene emissions (mean isoprene emission over the growing season 44 $\mu$g m$^{-2}$ h$^{-1}$), but sesquiterpenes (SQT) emissions were higher than MT emissions all the time. The main MTs emitted were $\alpha$-pinene, 1,8-cineol, myrcene, limonene and 3$\Delta$-carene. Of SQTs cadinene, $\beta$-cadinene and $\alpha$-farnesene had the major contribution. During early growing season SQT/MT emission rate ratio was ~10, but it became smaller as summer proceeded being only ~3 in July. Isoprene, MT and SQT emissions were exponentially dependent on temperature (correlation coefficients ($R^2$) 0.75, 0.66 and 0.52, respectively). Isoprene emission rates were also found to be exponentially correlated with the gross primary production of $CO_2$ ($R^2$=0.85 in July).

Even with the higher emissions from the wetland, ambient air concentrations of isoprene were on average >100, >10 and >6 times lower than MT concentrations in May, June and July, respectively. This indicates that wetland was not the only source affecting atmospheric concentrations at the site, but surrounding coniferous forests, which are high MT emitters, contribute as well. Daily mean MT concentrations had high negative exponential correlation ($R^2$=0.96) with daily mean ozone concentrations indicating that vegetation emissions can be a significant chemical sink of ozone at this sub-Arctic area.

## 1. Introduction

Many plants communicate by releasing massive amounts of biogenic volatile organic compounds (BVOCs) into the air. They can either attract pollinators, repel herbivores or respond to other stress. Globally, these biogenic emissions are estimated to reach 760 Tg C yr$^{-1}$ and chemically consist mainly of isoprene ($C_5H_8$), monoterpenes (MT, $C_{10}H_{16}$), sesquiterpenes (SQT, $C_{15}H_{24}$), methanol and acetone (Sindelarova et al., 2014). The terrestrial biogenic VOCs account for about 90 % of the emission total (Guenther et al., 1995).

The global climate change has been estimated to produce the greatest warming in the high Northern latitudes (IPCC, 2018). Many plant species in these ecosystems are at the northernmost limits of their existence and even small changes in, for example, temperature and hydrology may severely influence vegetation composition and thus biosphere-atmosphere interactions and trace gas fluxes. The warming is likely to increase the emissions of biogenic VOCs into the atmosphere. Most of these emitted compounds have high reactivity in the atmosphere and their lifetimes vary from minutes to hours. These VOCs are oxidized mainly by the hydroxyl radicals, ozone and nitrate radicals and their oxidation products affect the secondary organic aerosol formation (SOA) formation, ozone fluxes and oxidative capacity of the air having therefore strong effects on air quality and climate. Knowing the current biogenic emissions in a very pristine subarctic environment will be valuable later when assessing the impact of progressing climate change in the Arctic region.

Knowledge of the biogenic emissions in the subarctic/Arctic area is very limited, at least partly because emissions are assumed to be low there due to low temperatures, short summers, and low vegetation biomass. However, atmospheric reactions such as SOA and ozone formation/destruction are regional rather than global processes (Tunved et al., 2006). Therefore these relatively low emissions can have high impacts in this very pristine environment. In addition, Kramshoj et al. (2016) showed that temperature dependence of BVOC emissions in the Arctic can be significantly higher than in tropical ecosystems. Therefore, the predicted temperature change has a vast influence on the BVOC emissions and will cause a significant rise compared to the observations so far.

Wetlands cover an area of about 2% of the total land surface area of the world. Most of the wetlands are located in the boreal and tundra zones on the northern hemisphere (Archibold, 1995). Approximately 3.5 million km$^2$ of boreal and subarctic peatlands exist in Russia, Canada, the United States (Alaska), and Fennoscandia (Finland and the Scandinavian countries).

Northern wetlands are important sinks for carbon dioxide and sources of methane, but knowledge on their VOC emissions is very limited. Studies have shown, that northern wetlands are high isoprene emitters (Janson and Serves, 1998; Haapanala et al. 2006; Hellén et al., 2006; Ekberg et al., 2009; Holst et al., 2010), but very little is known about the physiological and environmental regulations of isoprene fluxes especially in colder subarctic environments and even less is known on the emissions of other VOCs. These other VOCs (e.g. mono- and sesquiterpenes) may have stronger effects on SOA formation even with lower emissions due to their higher SOA formation potentials compared to isoprene (Lee et al. 2006).

We have studied VOC emissions and their ambient concentrations at a Finnish sub-Arctic wetland (Lompolojänkkä), which is an open, nutrient-rich sedge fen (Lohila et al., 2015), and a part of the Pallas-Sodankylä Global Atmosphere Watch (GAW) station.

## 2. Experimental

### 2.1 Measurement site

The measurements were conducted in 2018 in a northern boreal fen, Lompolojänkkä, in the Pallas-Yllästunturi National Park, Finland. The Lompolojänkkä measurement site is an open, nutrient-rich sedge fen located in the aapa mire region of north-western Finland (67°59.832′N, 24°12.551′E, 269 m a.s.l.; Aurela *et al*. 2009, 2015; Lohila *et al*. 2010, 2015). The relatively dense vegetation layer is dominated by Dwarf Birches (*Betula nana*), Bogbeans (*Menyanthes trifoliate*), Downy willows (*Salix lapponum*) and sedges (*Carex spp*). The mean vegetation height on the fen is 40 cm. The moss cover on the ground is patchy (57% coverage), consisting mainly of peat mosses (*Sphagnum angustifolium, S. riparium and S. fallax*) and some brown mosses (*Warnstorfia exannulata*). This aapa mire is characterized by a relatively high water level, with almost the entire peat profile being water-saturated throughout the year. The maximum peat thickness is 3 m. The site is surrounded by coniferous forest, which consist mainly of Scots pine (Pinus sylvestris) (72% of the forest area) with some Norway spruce (Picea abies) (20%) and deciduous trees (mainly Betula pubescens, some Populus tremula and Salix spp.) (8%) (Lohila et al. 2015).

### 2.2. Emission measurements

The measurement setup consisted of a 60 x 60 x 25 cm fluorinated ethylene propylene (FEP) chamber, which was flushed with 4 L min$^{-1}$ of volatile organic compound (VOC) free air generated by a commercial catalytic converter (HPZA-7000, Parker Hannifin Corporation). Due to high reactivity of the studied compounds, their ambient air concentrations at this remote location are much lower than concentrations in our emission chamber. Mean concentrations in the chamber were 10 400 ng m$^{-3}$, 2 300 ng m$^{-3}$ and 57 000 ng m$^{-3}$ for SQTs, MTs and isoprene, respectively. Therefore we do not expect to create any artificial emission by using zero air. The thermal desorption - gas chromatograph - mass spectrometer (TD-GC-MS) was connected to the chamber via a 21 m long 4 mm (i.d.) tubing heated few degrees above the ambient temperature and pumped with 0.5 L min$^{-1}$ make up flow, followed by 2.5 m long, 1 mm (i.d.) tube, with a flow of 40 mL min$^{-1}$. All tubing was made of FEP. For in situ analysis the VOC samples were collected directly into a standard low flow cold trap filled with Tenax TA (50%) and Carbopack B (50%) of the TD-GC-MS for 30 min every two hour in May-June and every hour in July. The trap was kept at 20 °C during sampling to prevent water vapour present in the air from accumulating in the trap. Offline samples were taken in the end of July using Tenax TA/Carbopack B adsorbent tubes. Sampling flow was ~55 ml min$^{-1}$. Sampling time for tubes was 2 h during the day and 10 h during the night. Breakthrough of isoprene was possible during 10 h sampling. However, there were only three 10 h samples taken and they were over the nights, when the emissions were very low. Samples were analysed later in the laboratory of the Finnish Meteorological Institute.

Two custom-built data loggers were used during the campaign. From April to mid-July the set-up consisted of a thermistor (Philips KTY 80/110, Royal Philips Electronics, Amsterdam, Netherlands) for the temperature inside the chamber and a quantum sensor (LI-190SZ, LI-COR, Biosciences, Lincoln, USA) for the photosynthetically active radiation (PAR), measured just above the enclosure. In mid-July this data logger had technical problems, so a replacement was used. The new data-logger

used a LM35 (Texas Instruments, Dallas, USA) temperature sensor and a SQ-520 (Apogee Instruments Inc., Logan, USA) quantum sensor for PAR.

The temperature inside the soil chamber increased sometimes to >40 °C (9 out of 264 measurement points). This is clearly higher than ambient air temperature, which stayed all the time below 29 °C. Due to this heat stress some additional emissions may have been induced. However, low vegetation canopies in high latitude ecosystems have been observed to have canopy temperatures up to 15 °C warmer than the ambient air temperature during clear sky conditions (Rinnan et al., 2014). Highest differences between the air temperature measured at the height of 3 m and soil chamber temperature were observed during the hot summer days with clear sky conditions. Difference >15 °C was observe for 24 samples out of 264 samples. Mean difference was 4.6 °C. Due to this heat stress emission rates shown are expected to be overestimated during clear sky conditions, but this is not expected to affect emission potentials, which are normalized to 30°C. Sometimes heating can cause additional stress for example due to drought, but at this very moisture wetland this was not a problem.

Three different stainless steel frames located close to the measurement cabin in the middle of the fen were used for the emission measurements. Stainless steel frames were installed into the fen 8 months before starting the measurements. Vegetation at Frame 1 was dominated by forbs (*Menyanthes trifoliate*, *Comarum palustre*, *Equisetum fluviatile)* and different sedges and Sphagnum moss species (Fig. 1). Vegetation in Frame 2 consisted of two willow species (*Salix lapponum, Salix phylicifolia*), Purple marshlocks (*Comarum palustre*), and various sedge and sphagnum moss species. Vegetation in Frame 3 was dominated by different sedges and sphagnum moss species together with smaller amount of Dwarf birch (*Betula nana),* Downy willows *(Salix lapponum),* Purple marshlocks (*Comarum palustre)* and Horsetails *(Equisetum fluviatile)*.

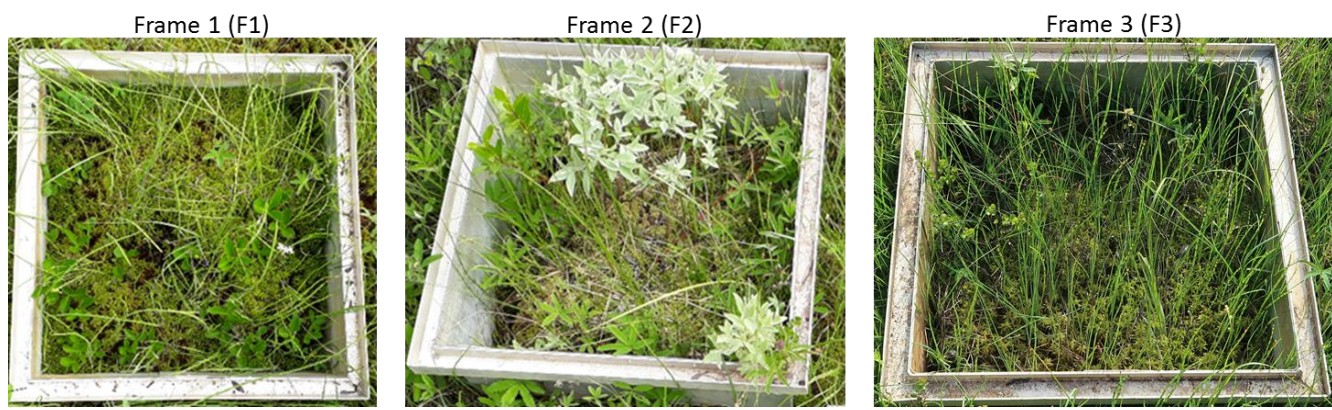

**Figure 1.**Vegetation cover of the measured frames in the beginning of July in 2018

### 2.3 Ambient air measurements

Ambient air samples were collected through a 21 m long FEP tubing (od. ¼ inch, id. 1/8 inch), which was heated to 5 °C higher than ambient temperature. The flow through the tube was 0.5 L min⁻¹. A stainless steel (grade 304) inlet line of 1 m was heated to 120 °C in order to destroy $O_3$. This $O_3$ removal method is described in detail by Hellén et al. (2012). Removal of $O_3$ from the inlet flow before collection of the sample is essential for avoiding losses of the very $O_3$-reactive compounds (e.g. β-caryophyllene). VOCs in a 40 mL min⁻¹ subsample were collected from the main inlet flow for 30 minutes every other

hour directly in the cold trap of the thermal-desorption unit. Samples were taken at the height of 1.5 m in the middle of the fen. In total, 273 ambient air samples were analyzed in May-July 2018.

## 2.4 GC-MS analysis

The in situ TD-GC-MS consisted of a thermal desorption unit (Turbo Matrix 650) followed by a gas chromatograph (Clarus 680) and a mass spectrometer (Clarus SQ8 T), all manufactured by Perkin-Elmer. An HP-5 column (60 m; i.d., 0.25 mm; film thickness, 1 µm from Agilent Technologies, Santa Clara, California, US) was used for the separation. The column was first heated from 50 to 150 °C at the rate of 4 °C min$^{-1}$ and then at the rate of 8 °C min-1 up to 250 °C, where it was kept for 5 min. Total time of the analysis was 42.50 min.

The instrument was located in a cabin in the middle of the fen. A five-point calibration was performed using liquid standards in methanol solutions. Standard solutions (5 µl) were injected onto adsorbent tubes and then flushed with nitrogen (80-100 ml min$^{-1}$) for 10 minutes to remove the methanol. The all studied MTs (α-pinene, camphene, β-pinene, 3Δ-carene, p-cymene, 1,8-cineol, limonene, myrcene, terpinolene ans linalool) and following SQTs were included in the calibration solutions: longicyclene, iso-longifolene, α-gurgunene, β-caryophyllene, β-farnesene and α-humulene. Unknown sesquiterpenes were tentatively identified based on the comparison of the mass spectra and retention indexes (RIs) with NIST mass spectral library (NIST/EPA/NIH Mass Spectral Library, version 2.0). RIs were calculated for all SQTs using RIs of known SQTs and MTs as reference. These tentatively identified SQTs were quantified using response factors of calibrated SQTs having the closest mass spectra resemblance. Isoprene was calibrated using gaseous standard (National Physical Laboratory, 32 VOC mix at 4 ppb level).

Offline samples taken on adsorbent tubes in the end of July were analyzed using the same method as for the in situ samples. TD-GC-MS used for the adsorbent tubes consisted of an automatic TD unit (TurboMatrix 650) connected to a GC (Clarus 600) coupled to a quadrupole MS (Clarus 600 T), all purchased from PerkinElmer Inc. (Waltham, MA, USA).

Detection limits for emission measurements were 0.002 - 0.055 µg m$^{-2}$ h$^{-1}$ and for ambient air measurements 0.3 – 5.1 pptv (See supplement Table S2). For calibrated compounds analytical precision calculated as a standard deviation of the calibration standards (N=6) varied between 1.5 and 5% and uncertainty estimated by following ACTRIS (Aerosol Clouds Trace gases Research InfraStructure) guidelines (ACTRIS, 2018) was between 17 and 25 %. Recoveries of the studied compounds from the used inlet (relative humidity 0% and 100%) and soil chamber (relative humidity 0%) were > 80%. However, linalool, bornylacetate and β-farnesene suffered from losses in the online mode of TD system at low humidity. In the soil emission samples humidity was high all the time and in the ambient air linalool, bornylacetate and β-farnesene concentrations were minor. Method validation has been described in detail by Helin et al. (2020).

## 2.5 Complementary data

The net ecosystem $CO_2$ exchange (NEE) between the biosphere and the atmosphere was measured by the eddy covariance method. The instrumentation consisted of a USA-1 (METEK) sonic anemometer and a closed-path LI-7000 (Li-Cor, Inc.) $CO_2/H_2O$ gas analyzer with a measurement height of 3 m (Aurela *et al.*, 2009). The gross primary production (GPP) was

determined from the NEE data utilizing standard partitioning procedures by estimating the respiration (R) from night-time NEE data and calculating GPP as the remainder of NEE and R (GPP=NEE-R) (e.g. Reichstein et al. 2005).

Supporting meteorological measurements, including air temperature (Vaisala, HMP), soil temperatures (PT100) at various levels, water table level (WTL) (PDCR1830), and photosynthetically active radiation (PAR) (Li-Cor, LI-190SZ), were collected by a Vaisala QLI-50 datalogger as 30-min averages.

## 3    Results and discussion

### 3.1  Terpenoid emissions and their environmental drivers.

Earlier studies have shown that isoprene is emitted from wetlands (Jansson and de Serves 1998; Haapanala et al. 2006; Hellén et al., 2006) and it turned out to be the most abundant compound in the current study also (Table 1 and Fig. 2). Isoprene emissions are dependent on temperature and light and therefore they are also connected to $CO_2$ gross primary production (GPP) as shown in Fig 3.  During the two first days in June, solar radiation was very high and also temperature in the enclosure increased, which is shown by the high light and temperature activity coefficients in Fig 3. This caused unexpectedly high isoprene emissions that clearly deviated from the general curve. The light and temperature activity coefficients (CT*CL) are described by Guenther et al. (1993) and are shown in Table 1 for each of the measurement periods separately.

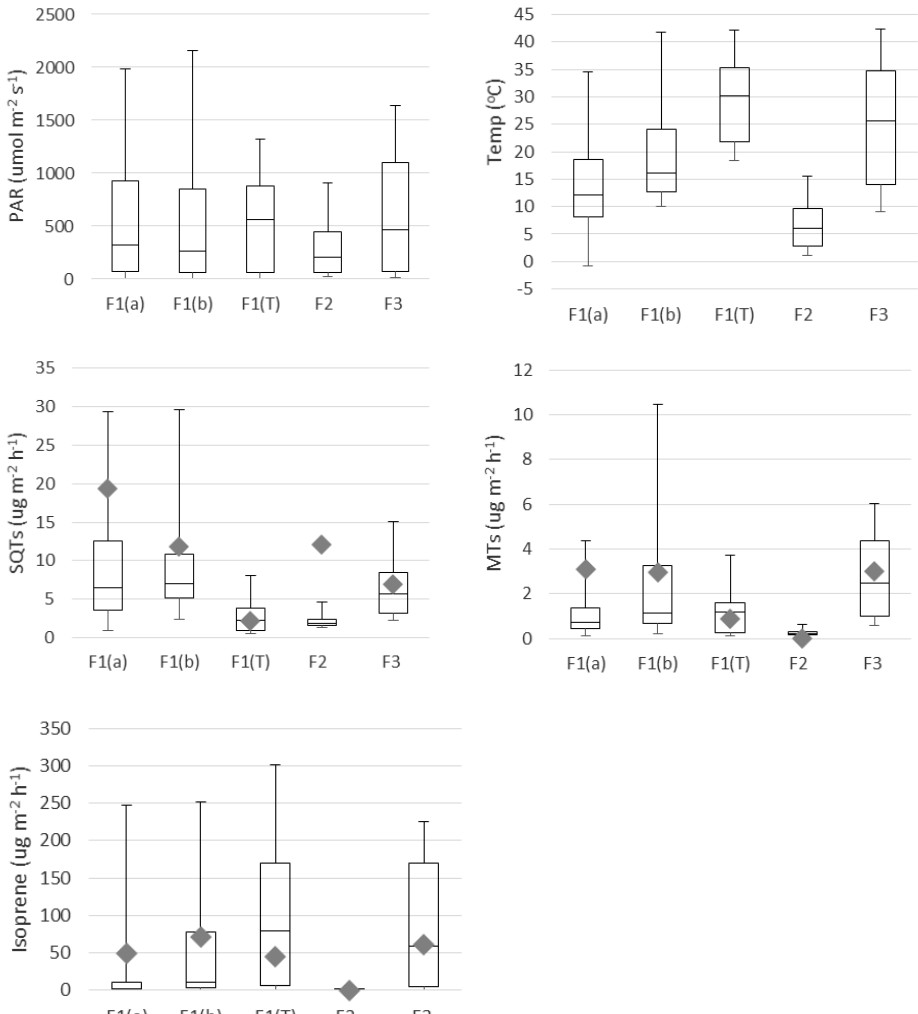

**Figure 2.** Mean box and whisker plots of photosynthetically active radiation (PAR), temperature (Temp) and emission rates of sesquiterpenes (SQTs), monoterpenes (MTs) and isoprene during the various emission measurement periods. The boxes represent first and third quartiles and the horizontal lines in the boxes the median values. The whiskers show the highest and lowest observations. Grey diamonds represent monthly mean emission potentials (T=30 ºC) and dots emission potential (T=30 ºC and PAR=1000 μmol m⁻² s⁻¹). F1(a)=frame 1 in 29/5 – 4/6, F2=frame 2 in 6/6 - 7/6, F1(b)= frame 1 in 4/7 – 10/7., F3=frame 12/7 – 13/7, F1(T)=tube samples from frame 1 in 30/7 -2/8.

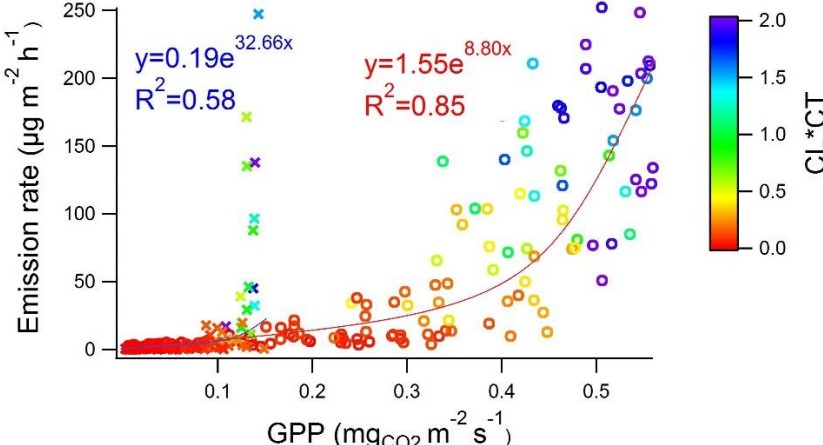

**Figure 3.** Isoprene emission rate vs. $CO_2$ gross primary production (GPP) in the beginning of summer (29/5 – 7/6, crosses and red fitting) and in July (4/7 – 13/7, dots and blue fitting) measured at Lompolojänkkä fen in summer 2018. Color corresponds to the strength of the light and temperature activation coefficient (CL*CT) in the range 0.0-2.0. $R^2$ values are from the corresponding transformed straight lines (ln(y), ln(x)).

**Table 1**. Mean chamber temperature (T), photosynthetically active radiation (PAR), temperature and light activity coefficient (CT*CL, Guenther et al., 1993) and emission rates (E) of monoterpenes (MTs), sesquiterpenes (SQTs) and isoprene during the various sampling periods. F1(a)=frame 1, F2=frame 2, F1(b)= frame 1, F3=frame 3, F1(T)=tube samples from frame 1. (<DL=below detection limit).

| Site | Snow | F1 | F1 | F1(T) | F2 | F3 |
|---|---|---|---|---|---|---|
| Sampling period | (25/4 - 8/5) | (29/5 - 4/6) | (4/7 - 10/7) | (30/.7 - 2/.8.) | (6/6 - 7/6) | (12/7 - 13/7) |
| Number of samples | | 73 | 91 | N=20 | N=10 | 19 |
| T [°C] | 3.5 | 14 | 20 | 29 | 7 | 26 |
| PAR [µmol m$^{-2}$ s$^{-1}$] | 390 | 540 | 560 | 530 | 290 | 720 |
| CT*CL | | 0.25 | 0.27 | 0.86 | 0.04 | 0.89 |
| $E_{SQTs}$ [µg m$^{-2}$ h$^{-1}$] | 0.004 | 8.8 | 8.7 | 2.5 | 2.1 | 7.3 |
| $E_{MTs}$ [µg m$^{-2}$ h$^{-1}$] | 0.2 | 1.1 | 2.1 | 0.9 | 0.2 | 3.0 |
| $E_{Isoprene}$ [µg m$^{-2}$ h$^{-1}$] | <DL | 18 | 49 | 90 | 0.2 | 82 |

MT emissions were generally less than 10% of the isoprene emissions, but SQT emissions were surprisingly high and exceeded MT emissions all the time (Table 1 and Fig. 2). To our knowledge, this is the first time SQT emissions have been measured from wetlands, although Faubert et al. (2010, 2011) detected emissions of SQTs from boreal peatland samples in laboratory experiments. Usually in the emissions from the vegetation MTs are several times higher than SQTs (e.g. Messina

et al. 2016), but here in the wetland emissions it was the opposite. During the early growing season, SQT emission rates were about ten times higher than MT emission rates but this difference became smaller since MT emissions increased as summer proceeded and reached their maximum in July, while SQT emission rates remained at about the same level until August. Late summer SQT emission rates were still about twice as high as MT. These high early summer SQT emissions could be due to compounds produced in soil and trapped in frozen soil and snow cover during winter. Aaltonen et al. (2012) measured VOCs in the snow cover in boreal forest in southern Finland and found both MTs and SQTs in snow with decreasing trend from soil surface to top layer of snow indicating soil as a source for terpenoids. These compounds can then be released to the air when snow and ground melt.

The emission patterns of both MTs and SQTs varied during the measurements. In Table 2, we show six most abundant compounds, both for SQTs and MTs. The most abundant MT was α-pinene in the beginning of measurements and it remained the most copious one until August, when 1,8-cineol exceeded α-pinene emissions. The most abundant SQT was cadinene replaced by β-cadinene in August (cadinenes are tentative identifications). Altogether, we identified 19 SQTs (supplementary Table S1), but we had standards for only 8 of them. The rest are identified and quantified as described in the methods section.

**Table 2.** The most abundant MTs and SQTs in the emissions at Lompolojänkkä fen during each measurement period. F1-3 indicates the measured frame.

| MTs: | α-pinene | β-pinene | 3Δ-carene | myrcene | 1,8-cineol | limonene | others |
|---|---|---|---|---|---|---|---|
| 29/5-4/6 (F1) | 59% | 3% | 7% | 2% | 9% | 5% | 14% |
| 4/7-10/7 (F1) | 40% | 2% | 9% | 4% | 16% | 7% | 21% |
| 30/7-2/8 (F1) | 10% | 1% | 1% | 9% | 35% | 10% | 35% |
| 6/6-7/6 (F2) | 23% | 3% | 4% | 25% | 21% | 0% | 23% |
| 12/7-13/7 (F3) | 29% | 7% | 5% | 5% | 21% | 8% | 25% |
| SQTs: | α-farnesene | cadinene | isosatovene | β-cadinene | α-muurolene | SQT2 | others |
| 29/5-4/6 (F1) | 12% | 46% | 2% | 13% | 3% | 3% | 21% |
| 4/7-10/7 (F1) | 13% | 45% | 3% | 16% | 3% | 2% | 18% |
| 30/7-2/8 (F1) | 1% | 27% | 2% | 56% | 2% | 5% | 7% |
| 6/6-7/6 (F2) | 27% | 32% | 3% | 17% | 5% | 1% | 16% |
| 12/7-13/7 (F3) | 15% | 33% | 8% | 13% | 6% | 9% | 16% |

Both MT and SQT emissions were dependent on temperature as shown in Fig. 4. The β-coefficient (as described by Guenther et al., 1993) for MTs was 0.08 which is close to 0.09 often observed for terrestrial plants. The coefficient was lower for SQT (0.05). Lower coefficients have been previously observed also for SQT emissions from a Norway spruce (Hakola et al., 2017). For individual MTs and SQTs β-coefficient varied between 0.05 and 0.14 and between 0.04 and 0.10, respectively (Supplement TableS3). The β-coefficient of isoprene was 0.18. Higher than this temperature sensitivity of isoprene has been observed at a dry Arctic heath by Kramshoj et al. (2016). In their studies warming of 3.1ºC caused 240% increase in isoprene

emissions. They speculate that the large response to warming may be linked to the effects of decreased soil moisture, which may limit the natural cooling evapotranspiration process, resulting in higher leaf temperatures at this dry Arctic site.

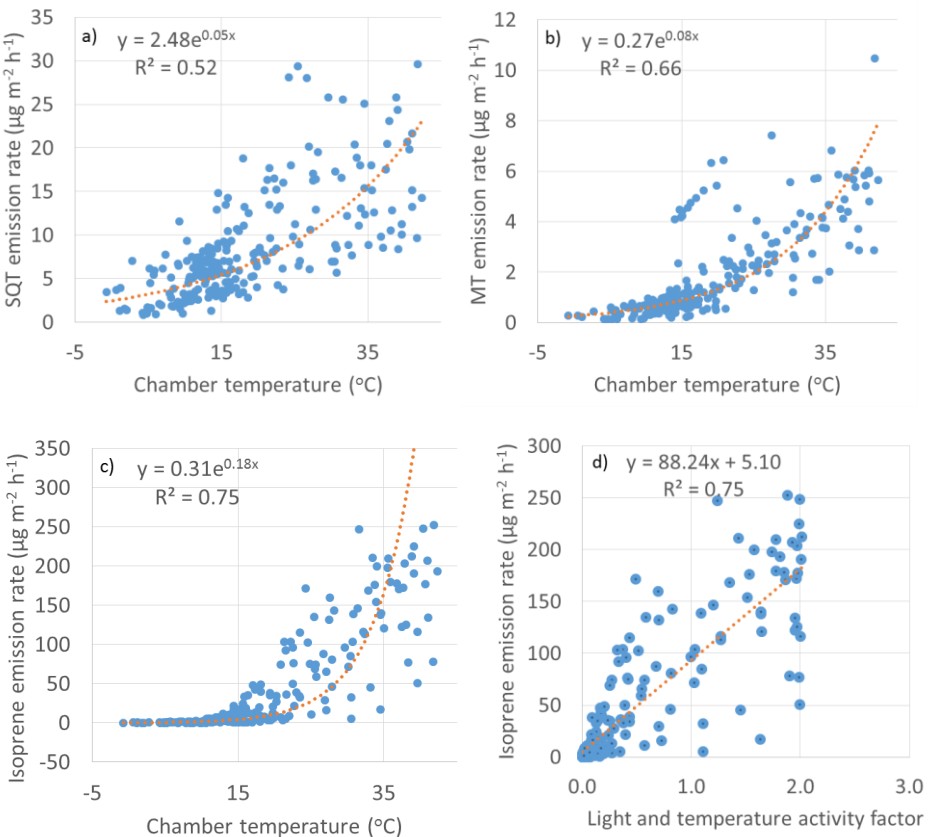

**Figure 4.** Exponential correlation of temperature with emission rates of a) SQTs, b) MTs and c) isoprene and d) linear correlation of light and activity coefficient (Guenther et al., 1993) with emission rates of isoprene. $R^2$ values for the plots a) - c) are from the corresponding transformed straight lines (ln(y), ln(x)).

SQT emissions were better correlated with soil temperature than chamber temperature in early summer, when soil temperature was below 15°C (Fig. 5) and vegetation was sparse (Fig. 6). This is additional proof for the hypothesis that SQTs are produced under the snow due to microbiological activity and released when snow is melting and soil thawing. Later on new growth affects the emissions, correlation with soil temperature weakens and correlation with chamber temperature increases. For MTs and especially for isoprene chamber temperature had better correlation also in early summer indicating stronger effect of new growth also then.

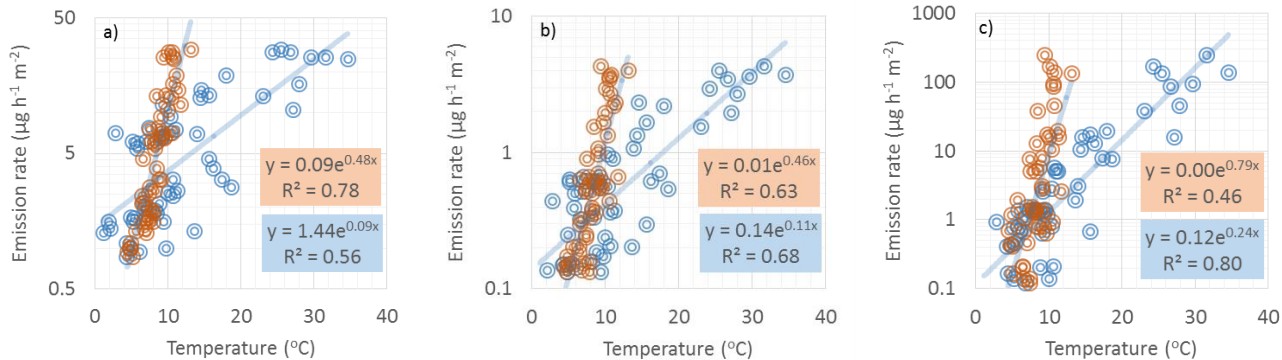

**Figure 5.** Correlation of a) SQTs, b) MTs and c) isoprene emission rates with soil (orange dots) and chamber (blue dots) temperatures during early growing season in 1/6/18 - 7/6/18. Soil temperature was measured 5 cm below the soil surface close to our emission chamber. $R^2$ values are from the corresponding transformed straight lines (ln(y), ln(x)).

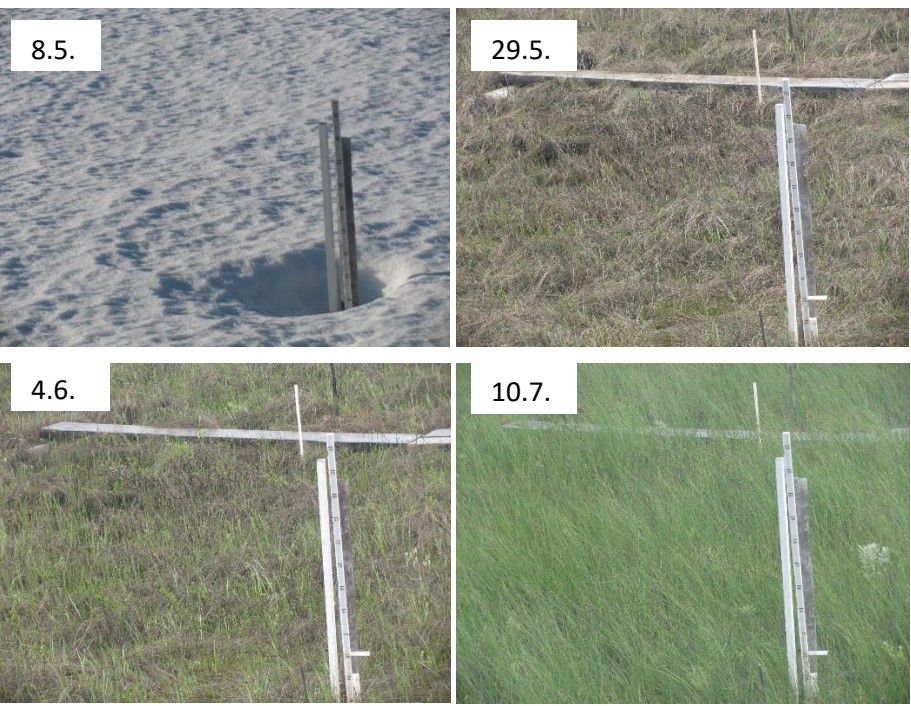

10     **Figure 6.** Variation of the vegetation cover at Lompolojänkkä fen over the growing season in 2018.

Mean diurnal variation of emissions of all terpenes followed the variations of temperature and PAR (Fig. 7). The isoprene emission is light dependent and the emissions were not detected during nights, but MTs and SQTs were emitted also then.

During night-time SQTs were clearly the most significant compound group emitted both in early and mid-summer. Also diel curves clearly show that SQT emissions were unexpectedly high in early summer (Fig. 7), which was also shown as high emission rates (Table 1).  Supplement Figure S1 show the mean diurnal variation of the individual SQTs and MTs for the same periods. Relative contribution of a SQT, β-cadinene, clearly increases during the daytime. Of the MTs daytime increase was observed for the 1,8-cineol, myrcene and limonene. Higher daytime contribution could indicate light dependent source of these compounds. Earlier light dependence of 1,8-cineol emissions has been observed in Scots pine emissions (Tarvainen et al., 2005).

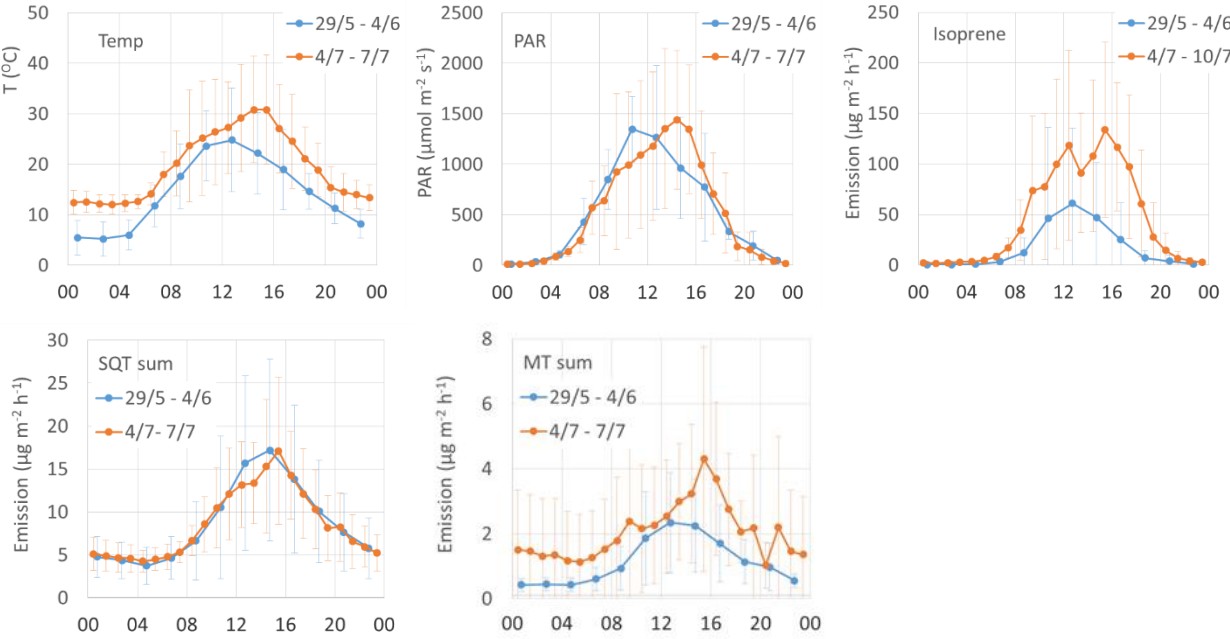

**Figure 7.** Diurnal variation of chamber temperature, PAR and emission rates (µg m$^{-2}$ h$^{-1}$) of isoprene, SQTs and MTs.

Mean emission potentials (at 30 °C) of SQTs (Fig. 2) decreased over the growing season while emission potentials of MTs stayed at the same level and isoprene potentials were slightly increasing indicating different source mechanisms for these compounds.

There is very little information available on the emissions of individual plant species growing at the Lompolojänkkä fen. Salix species are generally having high isoprene emissions (Iseprands et al. 1999, Kramshoj et al. 2016) and also wetland sedges are known to be strong isoprene emitters (Ekberg et al. 2009). Spaghnum moss is low emitter of both isoprene and MTs (Tiiva et al. 2009, Iseprands et al. 1999, Faubert et al. 2009). However, there is knowledge available on emissions of wetlands especially for the isoprene. The mean emission potential of isoprene (93 µg h$^{-1}$ m$^{-2}$ at 30 °C and 1000 µmol m$^{-2}$ s$^{-1}$) at the Lompolojänkkä fen was lower than measured earlier at a boreal fen in Southern Finland. In a study by Haapanala et al. (2009) emission potential measured with ecosystem scale flux measurements using relaxed eddy accumulation technique at  the

Southern boreal fen was on average 680 µg h$^{-1}$ m$^{-2}$ while emission potential measured using chambers was only 224 µg h$^{-1}$ m$^{-2}$ (Hellén et al., 2006). A reason for the high difference could be cold and rainy weather during the chamber measurements. If emissions rates with low (< 0.2) light and temperature activation values were used from ecosystem scale fluxes, emission potential was only 330 µg h$^{-1}$ m$^{-2}$. There have been high differences also in other earlier wetland studies. Janson and De Serves (1998) found high isoprene emission potentials (620 µg h$^{-1}$ m$^{-2}$) at wet southern boreal fen sites (flarks) while clearly lower potential (17 µg h$^{-1}$ m$^{-2}$) was measured for a dry site (hummock). In addition, Holst et al. (2010) measured ecosystem scale fluxes with a PTR-MS at a sub-Arctic wetland in Northern Sweden and found very high emission potential (1150 µg h$^{-1}$ m$^{-2}$) for isoprene.

Using a gradient method to measure BVOC fluxes above a boreal mixed forest very close to Lompolojänkkä site, Rinne et al. (2000) observed mean MT emission potential (at 30 °C) of 860 µg m$^{-2}$ h$^{-1}$, which is clearly higher than the mean emission potential of MTs (3 µg m$^{-2}$ h$^{-1}$) found for the Lompolojänkkä fen in our study. SQTs were not measured by Rinne et al. (2000), but in our study mean emission potential of SQTs from the Lompolojänkkä (11 µg m$^{-2}$ h$^{-1}$) was only ~1.5% of the emission potential of MTs found for the forest by Rinne et al. (2000). Mean isoprene fluxes (14.4 µg m$^{-2}$ h$^{-1}$) measured by Rinne et al. (2000) above this northern boreal forest were ~10 times lower than MT fluxes while in our study isoprene emission potential from the Lompolojänkkä fen was clearly higher (93 µg m$^{-2}$ h$^{-1}$) than for MTs. In an earlier study on a Sub-Arctic wetland in Sweden even much higher emission potential (1150 µg h$^{-1}$ m$^{-2}$) for isoprene has been observed (Holst et al., 2010). Since the wetlands cover large areas in the sub-Arctic areas, they can be a significant source of especially isoprene even compared to the emissions of the forests.

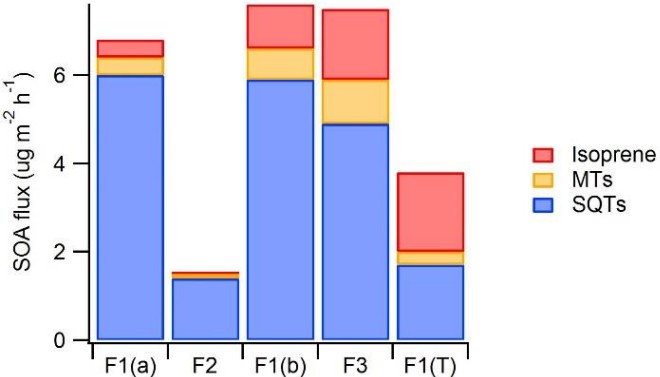

**Figure 8.** An estimate of the potential importance of emissions of different BVOC groups for SOA production. For calculating the SOA flux, BVOC emissions were weighed by the mean SOA yields found by Lee et al. (2006) in photo oxidation chamber studies (SQT 68%, MT 32% and isoprene 2%). F1(a)=frame 1 in 29/5 – 4/6, F2=frame 2 in 6/6 – 7/6, F1(b)= frame 1 in 4/7 – 10/7, F3=frame 12/7 – 13/7, F1(T)=tube samples from frame 1 in 30/7 – 2/8.

SQTs are very reactive and have higher SOA yields than MTs or isoprene (e.g. Lee et al., 2006) and therefore even lower emissions can have strong impacts on local chemistry and on aerosol formation and growth. To get a rough estimate of the

importance of different compound groups for SOA production, emission rates were weighed by the SOA yields found from the literature. Yields were obtained from a photo-oxidation chamber study by Lee et al. (2006) where the same set-up was used for studying SOA production of isoprene, MTs and SQTs. Yields for isoprene, α-pinene (MT) and β-caryophyllene (SQT), were 2%, 32% and 68%, respectively. In real environment there are lots of factors (e.g. meteorology, pre-existing aerosol load, oxidant, individual MTs and SQTs and NOx concentrations etc.) affecting the SOA production and therefore this gives just a rough estimate. However, results indicate that SQTs are the main SOA source at this wetland site especially during early growing season (Fig. 8). SQTs emitted by the forests are often oxidized already inside the canopy (Zhou et al., 2017) and oxidation products may be lost on canopy surfaces, but from the wetland all compounds are directly released to the atmosphere.

## 3.2 Snow emissions

Snow emissions over the Lompolojänkkä fen were studied between 25[th] of April and 8[th] of May using a similar chamber set up as for the fen emissions during summer (Table 1). In these measurements we detected low but clear emissions of MTs (mean $0.2\pm0.2$ µg m$^{-2}$ h$^{-1}$) especially during the sunny and warm days. The highest emission rate of MTs (1.5 µg m$^{-2}$ h$^{-1}$) was detected on 7[th] of May at 10:00-10:30 (UTC+2), when PAR was 1480 µmol m$^{-2}$ s$^{-1}$ and ambient temperature 6 °C. Main MT detected was α-pinene. Due to remote and clean location, even these low emissions may have strong impacts on the atmosphere. There is very little data on the VOC emissions on the snow cover. Low fluxes of MTs in the snowpack in boreal forest have been detected by Aaltonen et al. (2012) and Helmig et al. (2009). Also Swanson et al. (2005) have measured fluxes of VOCs through the snowpack, but they did not measure terpenoids.

## 3.3 Ambient air concentrations

Ambient air concentrations were measured at Lompolojänkkä in three periods over the growing season as shown in Table 3. Simultaneous emission and ambient air measurements were not possible since the same instrument was used for both.

Isoprene concentrations increased from May to July similar to their emission rates, but MT concentrations were higher than expected on the basis of May emissions from the fen. In addition to emissions also meteorology and chemical sink have huge effects on the concentrations levels (Zhou et al. 2017). In studies over grasslands dry deposition of isoprene and MTs have been detected as well (Spielmann et al. 2017, Bamberger et al. 2011) and in a study by Trowbridge et al. (2020) soil was shown to be a sink of isoprene in a deciduous hardwood forest, but due to strong emissions detected from the Lompolojänkkä wetland, soil uptake of isoprene is expected to be insignificant here. Since the lifetimes of MTs are a few hours, the area which affects the ambient air concentrations at Lompolojänkkä is large and within 10 km it is surrounded by the huge pine and spruce forests, which are known to emit mainly MTs (Rinne et al., 2000, Tarvainen et al. 2005, Hakola et al. 2017). Based on the flux measurements above the close by Norway spruce forest by Rinne et al. (2000), MT emission potential (30°C) from the forest is 286 times higher than in our wetland studies while isoprene emission potential is 10 times lower. Therefore the influence of the surrounding forests on ambient concentrations of MTs is expected to be high.

Based on the snow depth measurements at the site, there were still 22 cm of snow on 8[th] May, but on 12[th] of May no snow was present anymore (Fig. 6). Highest daily means for both MTs and SQTs were measured during this snow melting and soil thawing period (Fig. 9). Aaltonen et al., (2012) have shown that VOCs produced under the snow cover by for example microbes and decaying litter remain in the snow cover with quite high concentrations and are released to the air when snow is melting and ground thaws. Melting and thawing will most likely influence the surrounding forest emissions as well (Aalto et al. 2015). The MTs detected in ambient air were the same that were detected also in the emission measurements.

The SQTs measured in ambient air were not the same that were measured in the emissions of the fen, but the most abundant compounds in air were α-copaene and β-caryophyllene (Fig. 9). β-caryophyllene is very fast reacting compound, it's lifetime is only few minutes, and it was only minor fraction (<2%) of fen emissions. However, there can be differences between the measurement periods as shown in Table 2 and there were no parallel emission and ambient air measurements, but β-caryophyllene is also the main SQT in Scots pine and Norway spruce emissions (Tarvainen et al. 2005, Hakola et al., 2017). Therefore it is likely that surrounding Scots pine and Norway spruce forests can also be a SQT source. Other possible close by sources are mountain birches (Haapanala et al., 2009). Anyway this would need nearby and high β-caryophyllene emissions since its lifetime is only few minutes. Cadinene was the main SQT in the wetland emissions and it was detected in ambient air too. Main MT found in the ambient air, α-pinene, is the most important MT both in wetland and forest emissions (Rinne et al. 2000, Tarvainen et al. 2005, Hakola et al. 2017).

The atmospheric concentrations are a mixture of emissions from the wetland and the surrounding forest. This explains why the most abundant VOC class were the MTs and not e.g. isoprene or SQTs, which were higher emitted by the wetland. However, this comparison is not straight forward since isoprene is emitted only when daylight is available (Fig. 7), when also mixing layer of the atmosphere and therefore dilution effect is highest. MTs and SQTs are also emitted during the night and at least in a southern boreal forest highest concentrations of MTs and SQTs are often measured during the nights when the mixing layer is very shallow even though daytime emissions are much higher (Hellén et al. 2018).

Generally, ambient air concentrations of SQTs and MTs in Lompolojänkkä were at the same level as measured earlier in a boreal forest in Hyytiälä, Southern Finland (Kontkanen et al., 2016; Hellén et al., 2018). However, in May the concentrations at Lompolojänkkä were higher than in Hyytiälä boreal forest. The mean MT concentration measured by Kontkanen et al. (2016) in May 2006-2013 in Hyytiälä was ~150 pptv and SQT sum measured by Hellén et al. (2018) in May 2016 was 3 pptv while at Lompolojänkkä MT and SQT concentrations in May were more than two times higher 350 and 7 pptv, respectively (Table 2). Higher concentrations are not explained by the higher temperature. In Hyytiälä mean temperature in May 2006-2013 was 10 °C, which is the same as mean temperature during our measurements in May at Lompolojänkkä and during the SQT measurements in Hyytiälä in 2016 temperature was a bit higher 12.4 °C. However, concentrations of terpenes are known to vary a lot even at the same site depending on the exact measurement point (Liebmann et al., 2018)

Vertical ozone fluxes measured above forests are explained at least partly by stomatal uptake and non-stomatal sinks including chemical reactions with the BVOCs (Wolfe et al., 2011; Rannik et al., 2012). We found that daily mean MT and SQT concentrations had a negative exponential correlation with ozone concentrations as shown in Fig. 10. This indicates that BVOC emissions have a strong impact on ozone concentrations at this remote sub-Arctic site. However, the dependence was detected only in June and July. In May MT concentrations were highest and no correlation was observed. The reason could

be that during that time the main MT source impacting our measurements was very close by, perhaps in the snow cover on the fen as discussed earlier, while impact of VOCs on ozone concentrations is more regional phenomena. In June and July, due to low emissions at the fen, main source of MTs was expected to be more regional (e.g. surrounding forests). Lifetimes of MTs in relation to ozone reaction is few hours for most of the MTs (Hellén et al., 2018).

5 To estimate the effects of various BVOCs on local ozone chemistry we calculated ozone reactivities (R) of them by multiplying the measured concentrations with ozone reaction rate coefficients ($k_{O3}$) as in Eq 1.

$$R_i = [BVOC_i] \times k_{i,O3} \tag{1}$$

Used ozone reaction rate coefficients were obtained from the literature and are listed as supplement Table S4. This determines in an approximate manner the relative role of compounds or compound classes as an ozone sink. Of the measured compounds

10 SQTs had the highest (66%) contribution β-caryophyllene being clearly the most important SQT (Fig. 11). Isoprene had only <1% contribution. Of the MTs α-pinene and limonene had highest contributions (23% and 5% of the total BVOC $O_3$ reactivity, respectively). Contribution of other compounds was minor. Since SQTs are highly reactive (lifetime of β-caryophyllene ~2 minutes) their effect is very local while MTs with lifetime of a few hours have the more regional impact.

15 **Table 3.** Ambient air concentrations of sesquiterpenes (SQTs), monoterpenes (MTs) and isoprene during the ambient air measurement periods in May (8/5, 14/5 – 17/5 and 21/5 – 26/5), June (13/6 – 17/6) and July (1/7 – 3/7).

| | May N=142 | June N=103 | July N=28 |
|---|---|---|---|
| T [°C] | 10 | 11 | 16 |
| PAR [$\mu$mol m$^{-2}$ s$^{-1}$] | 420 | 450 | 610 |
| $O_3$ [ppbv] | 39 | 31 | 27 |
| SQTs [pptv] | 7 | 3 | 21 |
| MTs [pptv] | 350 | 58 | 560 |
| Isoprene [pptv] | 3 | 5 | 87 |

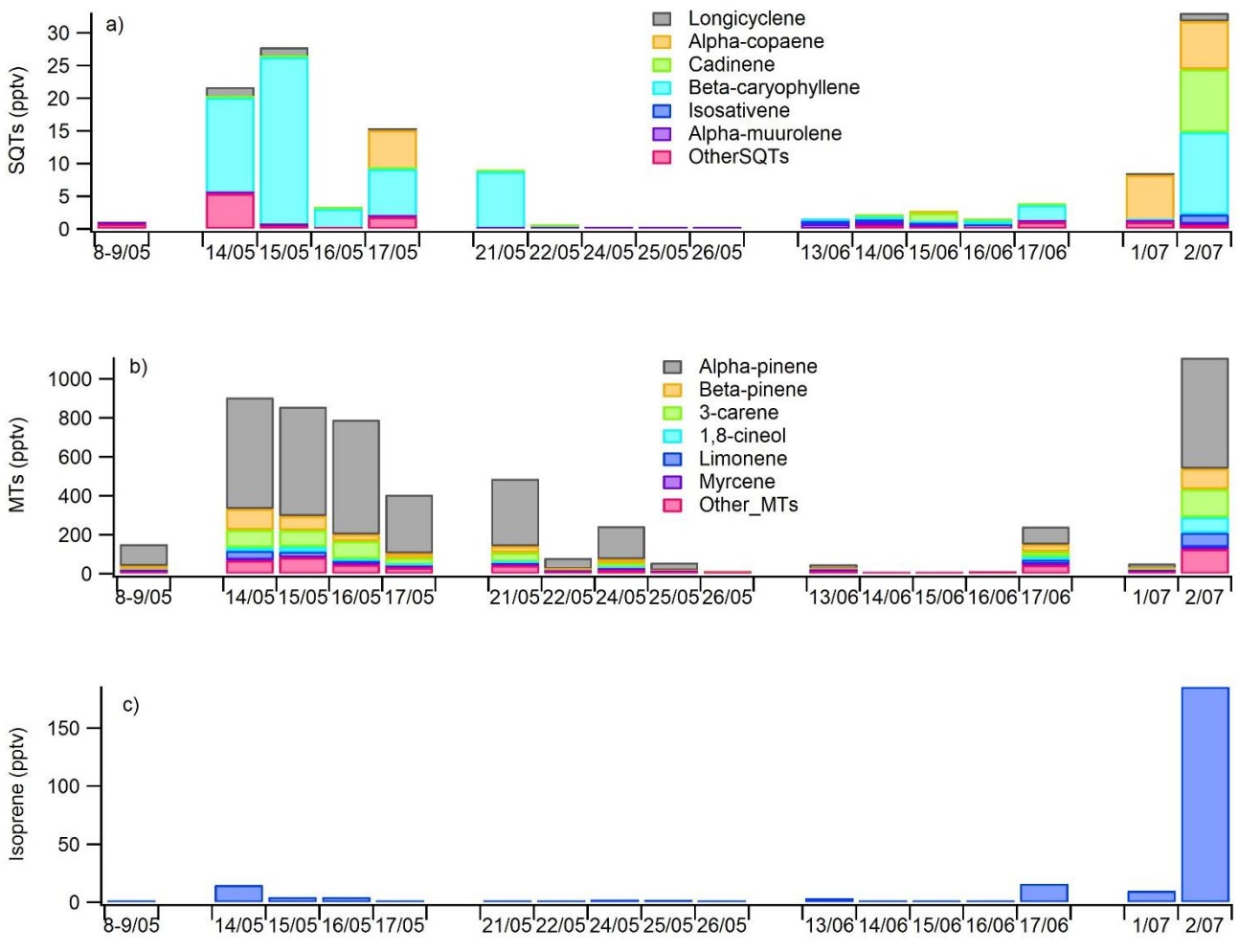

**Figure 9.** Daily mean concentrations of a) sesquiterpenes (SQTs), b) monoterpenes (MTs) and c) isoprene in the ambient air at Lompolojänkkä fen in 2018.

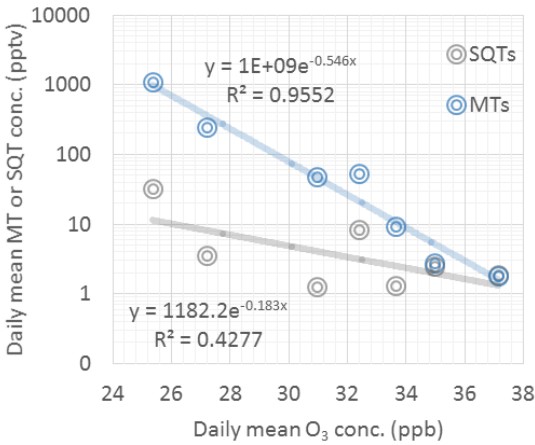

**Figure 10.** Dependence of daily mean ozone concentrations on daily mean monoterpene (MT) and sesquiterpene (SQT) concentrations during the ambient air measurements in June and July in 2018.

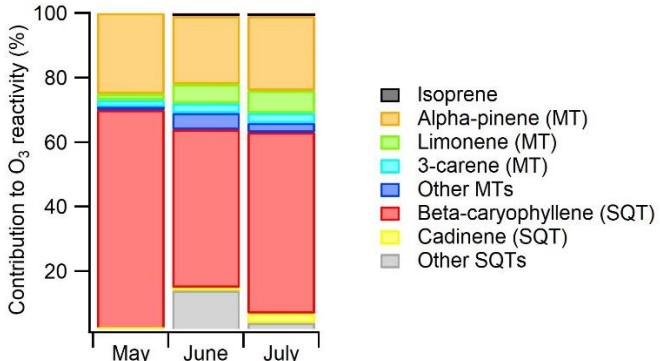

**Figure 11.** Contributions of various BVOCs to the ozone reactivity during the ambient air measurements at Lompolojänkkä fen in 2018. Used reaction rate coefficients are listed as supplement Table S4.

10  **4   Conclusions**

Emissions and ambient air concentrations of terpenoids were measured at the Lompolojänkkä fen in northern Finland in April-August 2018. Even though isoprene was the main terpenoid emitted from the fen and MT emissions were only 10 % of isoprene emissions, SQT emissions were several times higher than MT emissions, especially in early summer. Usually SQT emission from vegetation are several times lower than MT emissions. Emission rates of MTs were clearly lower than emission
15  rates measured earlier at the nearby spruce forest (Rinne et al. 2000), but isoprene emission rates from the fen were clearly higher.

Changes in light and temperature conditions explained most of the seasonal changes in isoprene emissions. Isoprene emission rates correlated well also with the gross primary production of $CO_2$. The sesquiterpene emissions were partly controlled by other factors, which was seen, for example, as decreasing emission potentials over the growing season. High SQT emissions during early summer suggested that the compounds trapped in frozen soil and snow cover during winter could be an important source of these compounds when the soil is thawing.

Comparison with the emissions from a close by forest indicated that the BVOC concentrations observed at the fen were not only affected by the fen emissions, but also by emissions from surrounding ecosystems. Nevertheless, the wetlands were shown to be a significant source of isoprene within this kind of sub-Arctic landscape. Furthermore, strong correlation of daily mean ambient air concentrations of MTs and SQTs with ozone concentrations indicated that BVOC emissions from these ecosystems have strong impact on the local ozone levels. Of the studied BVOCs β-caryophyllene had the main contribution on local ozone reactivity.

**Data availability**

GC-MS and complementary data used in this work are available from the authors upon request (heidi.hellen@fmi.fi).

**Author contributions**

H. Hellén designed and conducted the VOC measurements, performed the data analysis and led the writing of the manuscript. H. Hakola supervised the study, helped designing and setting up the measurement campaign and the commented on the manuscript. S.Schallhart, T.Tykkä and A. Praplan conducted the VOC measurements and data analysis and commented on the manuscript. M. Aurela and A. Lohila provided the complementary data, wrote their description and commented on the manuscript.

**Competing interests**

The authors declare that they have no conflict of interest.

**Acknowledgements**

The research was supported by the Academy research fellow project (Academy of Finland, project 275608) and Academy of Finland via the Center of Excellence in Atmospheric Sciences (grant no. 307331). The authors thank Timo Mäkelä and Juha Hatakka form Finnish Meteorological Institute for their help with setting up the measurements and Dr. Tarmo Virtanen from University of Helsinki for identifying the vegetation in the measured frames.

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
