# Peer review of "Sesquiterpenes dominate monoterpenes in Northern wetland emissions"

_Atmospheric Chemistry and Physics, 2019_

## Referee Comment (RC1) · Kolby Jardine (Referee) · 13 Feb 2020

Terpenoid emissions from high latitude wetland ecosystems have been poorly characterized, but important for understanding biological functions and atmospheric impacts as environmental drivers change.

The study by Hellén et al., 2020 provides valuable data on both biogenic emissions of terpenoid compounds and their ambient concentrations at a sub-Arctic wetland in Finland (Lompolojänkkä). Observations were made using an in-situ thermal desorption-GC-MS which is ideal for studying terpenoid volatile emissions as other sensitive atmospheric techniques (e.g. PTR-TOFMS) have a limited ability to distinguish isomers. The study confirms previous work that isoprene is one of the most abundant terpenod emit-

ted by the wetland ecosystem. Consistent with many ecosystems across the world, monoterpene emissions appeared at roughly 10% of isoprene emissions. However, while many ecosystems show significant sesquiterpene emissions, they are generally low relative to isoprene and monoterpenes. In contrast, in Lompolojänkkä, higher sesquiterpene emissions

Abstract: The abstract is highly qualitative, lacking quantitative data (with uncertainties). Simply stating that something is higher or lower is not acceptable, especially without statistical tests.

Which monoterpenes and sesquiterpenes were observed? How did their composition change with the growing season?

"Isoprene, MT and SQT emissions were dependent on temperature." What is the correlation? Positive, negative? Were there emissions at night? Are they light-dependent or independent?

"Isoprene emission rates were also found to be well-correlated with the gross primary production of $CO_2$. Even with the higher emissions from the wetland, ambient air concentrations of isoprene were clearly lower than MT concentrations. This indicates that wetland was not the only source affecting atmospheric concentrations at the site, but surrounding coniferous forests, which are high MT emitters, contribute as well." The authors need to consider uptake by the soil of isoprene, how can they rule this out? There seems to be a number of other possible scenarios to explain the findings.

"In May concentrations of SQTs and MTs at Lompolojänkkä were higher than in earlier boreal forest measurements in southern Finland. At that time the snow cover on the ground was melting and soil thawing and VOCs produced under the snow cover, e.g. by microbes and decaying litter, can be released to the air." Not clear which site the authors are referring to here and seems there is too much speculation.

"Daily mean MT concentrations were very highly negatively correlated with daily mean

ozone concentrations indicating that vegetation emissions can be a significant chemical sink of ozone at this sub-Arctic area." Please provide statistics!

This is fascinating, can the authors look further into this and estimate atmospheric terpene ozonolysis rates?

What is the relationship between the emission rates and the atmospheric concentrations?

Introduction: The introduction needs to be expanded to include background on the atmospheric and biological roles of terpenes. Especially the later as there is no mention of this here. I would like to see some sort of introduction on isoprene, monoterpenes, and sesquiterpenes.

Methods: Please cite references for the liquid standards in methanol. If only SQTs were present in the calibration solution, how were MTPs and isoprene calibrated?

Graphs: Please use different colors for alpha-pinene and other MTPs, they look the same!

Why did 2/07 show huge atmospheric concentrations of isoprene >150 ppb but the other days did not? That does not seem like a reasonable isoprene concentration.

Figure 7 with the diurnal patterns is beautiful data. I would be very interested to know if the composition of the terpenes changed across the day as previously observed in other sites.

---

## Referee Comment (RC2) · Anonymous Referee #2 · 10 Apr 2020

This study is a valuable contribution to the available observations for understanding BVOC emissions and atmospheric concentrations in northern wetlands. The manuscript is generally clear, concise and well written and the methods and uncertainties are well described.

As the authors indicate, the concentration data is difficult to interpret due to the influence of the nearby forest. It would be helpful if the authors could better describe the influence of the forest including species, expected BVOC fluxes, typical transport times of BVOC from the forest.

The title is misleading since this landscape is not a strong source of sesquiterpenes., The sesquiterpene emission factors reported for this study are similar to what model simulations (such as MEGAN) would predict for northern wetlands. Perhaps the title

could indicate that sesquiterpenes dominate monoterpenes, which is unusual. In any case, the abstract, text and conclusions should make it clear that the unusual MT/SQT ratio is because MT (and isoprene) are lower than most other landscapes, not because SQT are higher. Comments that sesquiterpenes are "surprisingly" high should be removed and could be replaced with a statement regarding the relative MT/SQT ratio.

As is discussed in the introduction, Kramshoj et al. and related work in an Arctic landscape in Greenland reports an isoprene temperature dependence that is much higher than in temperate landscapes. In contrast, Figure 4 shows that this northern wetland vegetation has an isoprene emission response that is similar to temperate vegetation. Please discuss the similarities and differences between this site and the Kramshoj site. Any insights on why the isoprene temperature response is so different?

Table 1 (and elsewhere in the manuscript): Please use a more standard format for the dates. Alternatively explain the format in the Table header or at least label them as dates.

Page 2, line 30-33: What is known about BVOC emissions from these various species in the fen ?

Page 3, line 7-9: What were the BVOC concentrations in the chamber?

Page 3, line 10: heated to what temperature?

Page 3, line 12: what was the size (mass of adsorbents) of the cold trap?

Page 3, line 14: what was the flow rate for the offline tube samples? Did the 10 hour samples exceed breakthrough volume for these tubes?

Page 3, line 25: The temperature difference is probably not as relevant as the absolute temperature. How realistic is it for these plants to have temperatures above 40C? Discuss the implications of heat stress impacts on these results.

Page 4, line 17: California is misspelled

Page 8, line 9: Since only frame #1 was sampled more than once, it would be clearer to show the seasonal data (i.e., the data for frame 1) and then separately show data for the other 2 frames. Otherwise it can appear all of the data are seasonal variations from the same location. All of the data could still probably go in one table or figure but just grouped differently.

Page 8, line 19: How does the temperature dependence vary for individual monoterpenes and sesquiterpenes?

Page 14, line 17-25: Which terpenes dominate the ozone uptake? This could be shown in a figure illustrating the contribution of each compound to total ozone reactivity (analogous to figure 8 for SOA).

---

## Author Comment (AC1) · 6 May 2020

Thank you for the very good comments! We have considered them and we have improved our manuscript based on them as explained in more detail here.

Terpenoid emissions from high latitude wetland ecosystems have been poorly characterized, but important for understanding biological functions and atmopsheric impacts as environmental drivers change.

The study by Hellén et al., 2020 provides valuable data on both biogenic emissions of terpenoid compounds and their ambient concentrations at a sub-Arctic wetland in Finland (Lompolojänkkä). Observations were made using an in-situ thermal desorption-GC-MS which is ideal for studying terpenoid volatile emissions as other sensitive atmospheric techniques (e.g. PTR-TOFMS) have a limited ability to distinguish isomers. The study confirms previous work that isoprene is one of the most abundant terpenoid emitted by the wetland ecosystem. Consistent with many ecosystems across the world,monoterpene emissions appeared at roughly 10% of isoprene emissions. However,while many ecosystems show significant sesquiterpene emissions, they are generally low relative to isoprene and monoterpenes. In contrast, in Lompolojänkkä, higher sesquiterpene emissions

Abstract: The abstract is highly qualitative, lacking quantitative data (with uncertainties). Simply stating that something is higher or lower is not acceptable, especially without statistical tests.

-more quantitative data has been added

Which monoterpenes and sesquiterpenes were observed? How did their composition change with the growing season?

-we added to the abstract 'The main MTs emitted were $\alpha$-pinene, 1,8-cineol, myrcene, limonene and 3$\Delta$-carene. Of the SQTs cadinene, $\beta$-cadinene and $\alpha$-farnesene had the major contribution.'

"Isoprene, MT and SQT emissions were dependent on temperature." What is the correlation? Positive, negative? Were there emissions at night? Are they light-dependent or independent?

-it was added that correlation with temperature was exponential with temperature R2 values being 0.75, 0.66 and 0.52 for isorepene, MTs and SQTs, reapectively.

"Isoprene emission rates were also found to be well-correlated with the gross primaryproduction of CO2.

-sentence was changed to 'Isoprene emission rates were also found to be exponentially correlated with the gross primary production of CO2 (R2=0.85 in July)'

Even with the higher emissions from the wetland, ambient airconcentrations of isoprene were clearly lower than MT concentrations. This indicatesthat wetland was not the only source affecting atmospheric concentrations at the site,but surrounding coniferous forests, which are high MT emitters, contribute as well."The authors need to consider uptake by the soil of isoprene, how can they rule this out? There seems to be a number of other possible scenarios to explain the findings.

-There are a few studies where deposition of both monoterpenes and isoprene over grasslands have been detected (Spielmann et al. 2017 and Bamberger et al. 2011). In addition Trowbridge et al. (2020) detected soil uptake of isoprene in a deciduous hardwood forest, but due to strong emissions detected from the Lompolojänkkä wetland, soil uptake of isoprene is expected to be insignificant here. Since the lifetimes of these compounds are few hours, concentrations of these compounds are low. However, the area which affects the concentrations is quite large. Vegetation around the site within 10 km is mainly spruce and pine forests, which are known to emit mainly monoterpenes. As discussed in the manuscript (section 3.3) emission potential of MTs from the nearby forest is 860 $\mu$g m$-2$ h$-1$ while emission potential of isoprene from the wetland is 93 $\mu$g m$-2$ h$-1$ and therefore these forests are expected to have huge effect on concentrations. More discussion on this was added to the manuscript into section 3.3.

"In May concentrations of SQTs and MTs at Lompolojänkkä were higher than in earlierboreal forest measurements in southern Finland. At that time the snow cover on theground was melting and soil thawing and VOCs produced under the snow cover, e.g.by microbes and decaying litter, can be released to the air." Not clear which site theauthors are referring to here and seems there is too much speculation.

-this too speculative sentence was removed from the abstract

"Daily mean MT concentrations were very highly negatively correlated with daily mean ozone concentrations indicating that vegetation emissions can be a significant chemical

sink of ozone at this sub-Arctic area." Please provide statistics!

-Correlation coefficients were added

This is fascinating, can the authors look further into this and estimate atmospheric terpene ozonolysis rates?

-a figure and discussion on ozone reactivity of the measured compounds were added to the manuscript into section 3.3

What is the relationship between the emission rates and the atmospheric concentrations?

-simultaneous measurements of emission rates and atmospheric concentrations were not possible since only one instrument was available

Introduction: The introduction needs to be expanded to include background on the atmospheric and biological roles of terpenes. Especially the later as there is no mentionof this here. I would like to see some sort of introduction on isoprene, monoterpenes, and sesquiterpenes.

-Introduction was expanded

Methods: Please cite references for the liquid standards in methanol. If only SQTs were present in the calibration solution, how were MTPs and isoprene calibrated?

-more details on the calibration was added into section 2.4

Graphs: Please use different colors for alpha-pinene and other MTPs, they look the same!

-colors have been changed

Why did 2/07 show huge atmospheric concentrations of isoprene >150 ppb but theother days did not? That does not seem like a reasonable isoprene concentration.

-The value is 150 pptv and not 150 ppbv. The higher value is due to higher temperature

[Figure]

and higher PAR. Due to exponential dependence this can have a huge effect.

Figure 7 with the diurnal patterns is beautiful data. I would be very interested to knowif the composition of the terpenes changed across the day as previously observed inother sites.

-Thank you for the very valuable comment. Changes in terpene composition was observed and additional figure (S1) was added to the supplement. Discussion on this was added to the manuscript: 'Supplement Figure S1 show the mean diurnal variation of the individual SQTs and MTs for the same periods. Relative contribution of a SQT, $\beta$-cadinene, clearly increases during the daytime. Of the MTs daytime increase was observed for the 1,8-cineol, myrcene and limonene. Higher daytime contribution indicates light dependent source of these compounds. Earlier light dependence of 1,8-cineol emissions has been observed in Scots pine emissions (Tarvainen et al., 2005).'

---

## Author Comment (AC2) · 6 May 2020

Thank you for the very good comments. We have considered them and we have improved our manuscript based on them as explained in more detail here.

This study is a valuable contribution to the available observations for understanding BVOC emissions and atmospheric concentrations in northern wetlands. Themanuscript is generally clear, concise and well written and the methods and uncertainties are well described.

As the authors indicate, the concentration data is difficult to interpret due to the influence of the nearby forest. It would be helpful if the authors could better describe theinfluence of the forest including species, expected BVOC fluxes, typical transport

times of BVOC from the forest.

-We added description of the forests surrounding the site into the section 2.1, but there is very little knowledge on the emissions of this kind of sub-Arctic forests. As discussed in the manuscript, emissions of close by Norway spruce forest have been measured by Rinne et al. (2000). However, their data set was very small and sesquiterpenes were not studied. Tarvainen et al. show some results of northern Scots pine emissions and Hakola et al. (2017) have measured emissions of a boreal Norway spruce. Discussion on these emissions were added to the manuscript into section 3.3. In addition, Haapanala et al. (2009) have measured emissions of mountain birches and this was added to the manuscript as well.

The title is misleading since this landscape is not a strong source of sesquiterpenes.,The sesquiterpene emission factors reported for this study are similar to what modelsimulations (such as MEGAN) would predict for northern wetlands. Perhaps the title could indicate that sesquiterpenes dominate monoterpenes, which is unusual. In anycase, the abstract, text and conclusions should make it clear that the unusual MT/SQTratio is because MT (and isoprene) are lower than most other landscapes, not because SQT are higher. Comments that sesquiterpenes are "surprisingly" high should be re-moved and could be replaced with a statement regarding the relative MT/SQT ratio.

-title has been changed to 'Sesquiterpenes dominate monoterpenes in Northern wetland emissions'

-Comments on unusual MT/SQT ratio was added and comments on 'suprisingly high' has been removed and ratio of MTs and SQTs are discussed

As is discussed in the introduction, Kramshoj et al. and related work in an Arctic landscape in Greenland reports an isoprene temperature dependence that is much higher than in temperate landscapes. In contrast, Figure 4 shows that this northernwetland vegetation has an isoprene emission response that is similar to temperatevegetation.

Please discuss the similarities and differences between this site and theKramshoj site. Any insights on why the isoprene temperature response is so different?

-discussion on results of Kramshoj et al. (2016) has been added to the manuscript

Table 1 (and elsewhere in the manuscript): Please use a more standard format for thedates. Alternatively explain the format in the Table header or at least label them asdates.

-Dates have been corrected

Page 2, line 30-33: What is known about BVOC emissions from these various species in the fen?

-There is very little information available on the emissions of these individual species. Salix species are generally having high isoprene emissions (Iseprands et al. 1999, Kramshoj et al. 2016) and wetland sedges are known to be strong isoprene emitters as well (Ekberg et al. 2009).Spaghnum moss is low emitter of both isoprene and MTs (Tiiva et al. 2009, Iseprands et al. 1999, Faubert et al. 2009). This was added to the manuscript into section 3.1.

Page 3, line 7-9: What were the BVOC concentrations in the chamber?

-mean concentrations of SQTs, MTs and isoprene were added.

Page 3, line 10: heated to what temperature?

-we added 'heated few degrees above the ambient temperature'.

Page 3, line 12: what was the size (mass of adsorbents) of the cold trap?

-excact amounts of adsorbents are not known, but it was added to the mansuscript that we used standard low flow cold trap of Perkin Elmer filled with Tenax TA (50%) and Carbopack B (50%).

Page 3, line 14: what was the flow rate for the offline tube samples? Did the 10 hour

samples exceed breakthrough volume for these tubes?

-The flow was ~55 ml/min. This was added to the manuscript. It is possible that even when using stronger adsorbent (Carbopack B) as a back up, most volatile isoprene suffered from the breakthrough during 10-hour samples. However, these samples were taken during the nighttime when emissions were low compare to the daytime emissions. These three points were above the general temperature dependence curve, which indicates that breakthrough was not significant. Comment on possible breakthrough was also added to the manuscript.

Page 3, line 25: The temperature difference is probably not as relevant as the absolute temperature. How realistic is it for these plants to have temperatures above 40C?Discuss the implications of heat stress impacts on these results.

-It was added that due to this heat stress emission rates shown are expected to be overestimated during clear sky conditions, but this is not expected to affect emission potentials, which are normalized to 30oC.

Page 4, line 17: California is misspelled

-Corrected Page 8, line 9: Since only frame #1 was sampled more than once, it would be clearerto show the seasonal data (i.e., the data for frame 1) and then separately show datafor the other 2 frames. Otherwise it can appear all of the data are seasonal variationsfrom the same location. All of the data could still probably go in one table or figure butjust grouped differently. -Measurement periods were grouped differently as suggested by the reviewer. Page 8, line 19: How does the temperature dependence vary for individual monoter-penes and sesquiterpenes? -information on the temperature dependence of individual terpenes were added to the manuscript and as a supplement table S3. Page 14, line 17-25: Which terpenes dominate the ozone uptake? This could be shownin a figure illustrating the contribution of each compound to total ozone reactivity (anal-ogous to figure 8 for SOA). -very good idea. A figure and discussion on this was added to the manuscript into section 3.3